# Local Unsupervised Learning for Image Analysis

**Leopold Grinberg**
IBM Research
lgrinbe@us.ibm.com

**John Hopfield**
Princeton Neuroscience Institute
Princeton University
hopfield@princeton.edu

**Dmitry Krotov**
MIT-IBM Watson AI Lab
IBM Research
krotov@ibm.com

## Abstract

We use a recently proposed biologically plausible local unsupervised training algorithm (Krotov & Hopfield, PNAS 2019) for learning convolutional filters from CIFAR-10 images. These filters combined with patch normalization and very steep non-linearities result in a good classification accuracy for shallow networks trained locally, as opposed to end-to-end. The filters learned by our algorithm contain both orientation selective units and unoriented color units, resembling the responses of pyramidal neurons located in the cytochrome oxidase "interblob" and "blob" regions in the primary visual cortex of primates. It is shown that convolutional networks with patch normalization significantly outperform standard convolutional networks on the task of recovering the original classes when shadows are superimposed on top of standard CIFAR-10 images. Patch normalization approximates the retinal adaptation to the mean light intensity, important for human vision. All these results taken together suggest a possibility that local unsupervised training might be a useful tool for learning general representations (without specifying the task) directly from unlabeled data.

## 1 Introduction

Local learning, motivated by Hebbian plasticity, has the following conceptual advantages compared to end-to-end training using a backpropagation algorithm. First, it only uses the activities of pre- and post-synaptic cells to learn the strength of a given synapse. This makes it possible to keep in the compute device memory only a small subset of all the neuron's activities, which is memory efficient. Second, it is naturally suitable for unsupervised learning tasks, when the labels are not available. Third, it is known to exist in real biological neural networks.

A variety of local learning algorithms relying only on bottom-up propagation of the information in neural networks have been discussed in the literature [1, 2, 3, 4, 5, 6, 7, 8]. A recent paper [8], for example, proposed a learning algorithm that is local and unsupervised in the first layer. It manages to learn useful early features necessary to achieve a good generalization performance, in line with networks trained end-to-end on simple machine learning benchmarks. The limitation of this study is that the proposed algorithms were tested only in fully connected networks, only on pixel permutation invariant tasks, and only on very simple datasets: pixel permutation invariant MNIST and CIFAR-10. An additional limitation of study [8], which is not addressed in the present work, is that they studied neural networks with only one hidden layer.

Our main contributions are the following. Based on the open source implementation of the algorithm proposed in [8], we designed an unsupervised learning algorithm for networks with *local* connectivity. We wrote a fast CUDA library that allows us to quickly learn weights of the convolutional filters at scale. We propose a modification to the standard convolutional layers, which includes patch normalization and very steep non-linearities in the activation functions. This allows us to match the performance of networks of similar size and architecture trained using the backpropagation algorithm end-to-end on CIFAR-10. The usefulness of patch normalization is illustrated by designing an artificial test set from CIFAR-10 images that are dimmed by shadows. The network with patch

NeurIPS 2019 workshop "Real Neurons & Hidden Units", Vancouver, Canada.

normalization outperforms the standard convolutional network by a large margin on this task. We also note that the filters learned by our algorithm show a well pronounced separation between color-sensitive cells and orientation-selective cells. This computational aspect of the algorithm matches nicely with a similar separation between the stimulus specificity of the responses of neurons in blob and interblob pathways, known to exist in the V1 area of the visual cortex.

The idea of using a convolutional network that is trained using only bottom-up learning signals is closely related to neocognitron [1]. The specific learning algorithm and the non-linear activation functions, discussed in the next section, are however different from the neocognitron learning algorithm [9] and activation functions.

## 2   Learning Algorithm and Network Architecture

During training, each input image is cut into small patches of size $W \times W \times 3$. The resulting set of patches $v_i^A$ (index $i$ enumerates pixels and RGB channels, index $A$ enumerates different patches) is shuffled at each epoch and is organized into minibatches that are presented to the learning algorithm. The learning algorithm uses weights $M_{\mu i}$, which is a matrix of $K$ channels by $N = W \cdot W \cdot 3$ visible units, that are initialized from a standard normal distribution and iteratively updated according to the following learning rule [8]

$$\Delta M_{\mu i} = \varepsilon \sum_{A \in \text{minibatch}} g\Big[\text{Rank}\Big(\sum_j M_{\mu j} v_j^A\Big)\Big]\Big[v_i^A - \Big(\sum_k M_{\mu k} v_k^A\Big)M_{\mu i}\Big] \tag{1}$$

where $\varepsilon$ is the learning rate. The activation function $g(\cdot)$ is equal to one for the strongest driven channel and is equal to a small negative constant for the channel that has a rank $m$ in the activations

$$g(i) = \left\{ \begin{array}{cl} 1, & \text{if } i = 1 \\ -\Delta, & \text{if } i = m \\ 0, & \text{otherwise} \end{array} \right. \tag{2}$$

Ranking is done for each element of the minibatch separately. The weights are updated after each minibatch for a certain number of epochs, until each row of the weight matrix converges to a unit vector (which is guaranteed for small values of $\Delta$, see [8]).

The resulting matrix $M_{\mu i}$ is used as weights of the convolutional filters with two important modifications. Frist, each patch $v_i$ of the image is normalized to be a unit vector before taking the dot product with the weight matrix. Given that the rows of the weight matrix themselves are unit vectors, the dot product between the weight and the patch is a cosine of the similarity between the two. Thus, $\sum_i M_{\mu i} v_i \in [-1, 1]$. Second, the result of the dot product is passed through a very steep non-linearity - rectified power function [10, 11]

$$f(x) = \Big[\text{ReLU}(x)\Big]^n \tag{3}$$

where the power $n$ is a hyperparameter of that layer, which controls sparseness. We call these slightly unusual convolutional layers NNL-CONV layers (normalized non-linear convolutional layers), in order to distinguish them from the standard ones, denoted CONV in this paper. The standard CONV layers do not use per-patch normalization and use ReLU as an activation function. In the following sections we also use standard max-pooling layers and standard fully connected layers with softmax activation function for the classifier.

## 3   Evaluation of the model on CIFAR-10 dataset

The proposed algorithm was applied to images from CIFAR-10 dataset to learn the weights of the NNL-CONV filters. They are shown in Fig.1. Each small square corresponds to a different channel (hidden unit) and is shown by projecting its corresponding weights into the image plane. The connection strength to each pixel has 3 components corresponding to the RGB color image. Thus, black color corresponds to synaptic weights that are either equal to zero, or negative; white color corresponds to weights that are large and have approximately equal values in all three RGB channels; blue color corresponds to weights that have large weights connected to blue neurons, and small or zero weights connected to red and green neurons, etc. As is clear from this figure, the resulting weights show a diversity of features of the images, including line detectors, color detectors, and detectors of more complicated shapes.

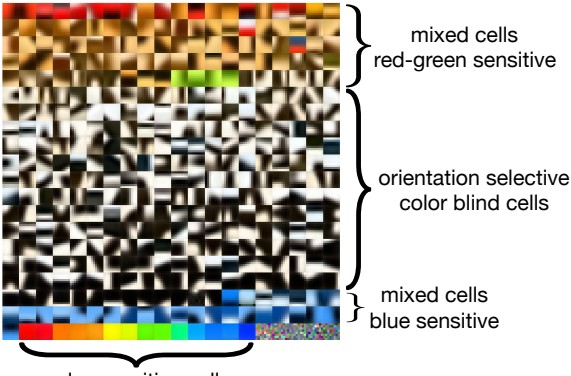

Figure 1: Filters of size $W = 8$ pixels, trained using the proposed algorithm. The filters are ordered to emphasize the differences between several groups of cells. The last five units in the last row did not learn any useful representation. They can be deleted from the network.

Guided by these examples, one can make the following qualitative observations. First, the majority of the hidden units are black and white, having no preference for R,G, or B color. These units respond strongly to oriented edges, corners, bar ends, spots of light, more complicated shapes, etc. Second, there is a significant presence of hidden units detecting color. Those neurons tend to have a smaller preference for orientation selectivity, compared to black and white units.

In order to test the quality of these learned filters with respect to the generalization performance, they were used as the weights of a simple network with one NNL-CONV layer. Consider for example

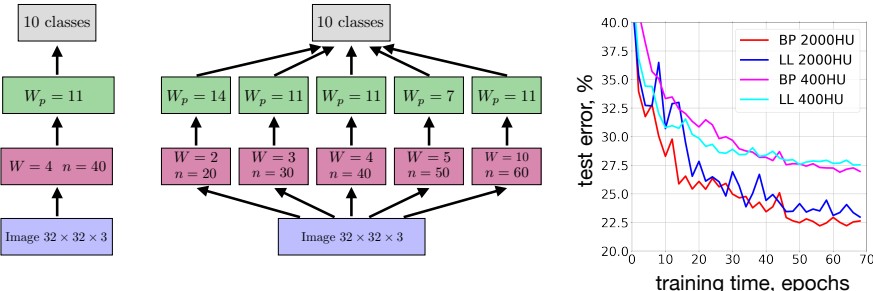

Figure 2: (Left) A simple architecture with an image projected to the NNL-CONV layer (shown in red), max pooling layer (shown in green), and a softmax classifier (gray); Parameters: convolutional window size $W = 4$ pixels; the power of the activation function is $n = 40$; number of channels $K = 400$, pooling window size $W_p = 11$ pixels (all the hyperparameters were determined on the validation set). (Middle) An architecture with blocks of different $W$ and $n$. (Right) The test set error of the two networks shown in the left and middle panels together with their end-to-end counterparts as a function of supervised training time.

the architecture shown in Fig. 2 (left). In this architecture a NNL-CONV layer is followed by a max pooling layer and then a fully connected softmax classifier. The weights of the NNL-CONV layer were fixed, and given by the output of the proposed algorithm. The max-pooling layer does not have any trainable weights. The weights of the top layer were trained using the gradient decent based optimization (Adam optimizer with the cross-entropy loss function). The accuracy of this network was compared with the accuracy of the network of the same capacity, with NNL-CONV layers replaced by the standard CONV layers, trained end-to-end using Adam optimizer. The results are shown in Fig. 2 (right). Here one can see how the errors on the held-out test set decrease as training progresses. Training time here refers to the training of the top layer classifier only in the case of the NNL-CONV network, and training of all the layers (convolutional layer and the classifier) in the case of the standard CONV network trained end-to-end. For the simple network on the left the error of the locally trained network is $27.80\%$, the error of the network trained end-to-end is $27.11\%$.

In order to achieve a better test accuracy it also helps to organize the NNL-CONV layer as a sequence of blocks with various sizes of the window $W$, like in Fig. 2 (middle). This helps to detect features of different scales. As above, the performance of this network trained using the proposed local learning algorithm is compared with the performance of a similar size network trained end-to-end. The results are shown in Fig. 2 (right). Both networks show errors $\approx 23\%$.

The main conclusion here is that the networks with filters obtained using the local unsupervised algorithm achieve almost the same accuracy as the networks trained end-to-end. This result is at odds with the common belief that the first layer feature detectors should be crafted specifically to solve the narrow task specified by the top layer classifier. Instead, this suggests that a general set of weights of the first layer can be learned using the local bottom-up unsupervised training, and that it is sufficient to communicate the task only to the top layer classifier, without adjusting the first layer weights.

Filters that look qualitatively similar to ours were previously obtained using several unsupervised learning techniques. Examples include sparse coding [2], unsupervised K-means clustering [12]. Competitive results (classification error on CIFAR-10 dataset $\approx 20\%$) using similar evaluation (unsupervised pre-training of the lower layers followed by the supervised training of the classifier), but different learning algorithm, were previously reported in [12, 6].

We also acknowledge that higher accuracies are common on CIFAR-10 dataset, but they require deeper architectures, use of regularizers (e.g. dropout), adversarial training, various forms of data augmentation. We compare the results of our algorithm with supervised training using plain back-propagation with Adam optimizer.

## 4 Color Sensitivity, Orientation Selectivity and Cytochrome Oxidase Stain

An interesting anatomical feature of the primary visual cortex is revealed when cells are stained with a cytochrome oxidase enzyme. This staining shows a pattern of blobs and interblob regions. In a famous set of experiments [13] it was discovered that the cells in the interblob regions are highly orientation selective and respond to luminance rather than to color, while the neurons inside the blobs respond to colors and are insensitive to orientation.

There are many important details and subtleties of these experiments that are not discussed in this paper, e.g. the neocortical visual areas of the primate brain have six anatomical layers many clearly divided into sublayers; neurons in different layers have different response properties; the response properties also vary smoothly within a layer. These facts limit the usefulness of describing how a "typical" pyramidal cell in V1 responds, especially to strong natural stimuli. However, the literature supports the qualitative assertion that there exist a segregation of orientation selective and color processing cells [13]. A segregation resembling the one discussed above can be seen in the learned filters shown in Fig. 1, which can be interpreted as preferred stimuli of the hidden neurons.

## 5 Patch Normalization, Retinal Adaptation, and Shadows

Natural scenes can have a several thousand fold range of light intensity variation [14]. At the same time, an 8-bit digital camera has only 256 possible intensity values. In order to cope with this huge variation of light intensities two separate systems exist in biological vision: a global control based on changing the size of the pupil, and a local adaptation on the retina. The latter, being the dominant one, enables the retina to have a good signal to noise ratio in both sun and shadow regions of a visual scene [14]. Patch normalization, which is essential for a good performance of our algorithm, can be

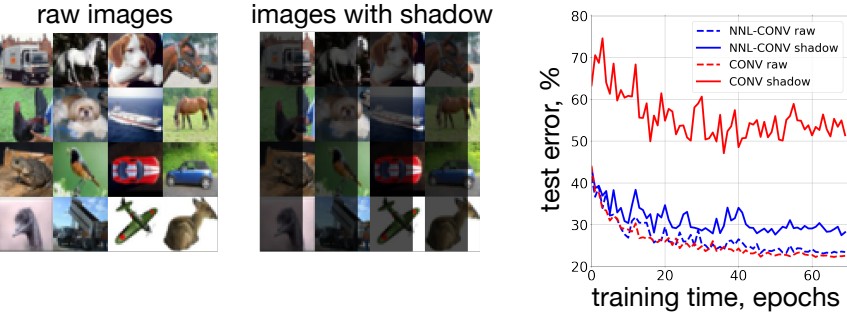

Figure 3: (Left) Randomly selected images from CIFAR-10 dataset. (Middle) Same images, part of each image is multiplied by an arbitrary parameter $0.3$ pixelwise. This imitates a shadow. (Right) Errors on the test set for the model with local patch normalization (blue) and a standard convolutional network (red). Raw images - dashed lines, shadowed images - solid lines.

thought of as a mathematical formalization of the local circuitry on the retina that is responsible for this adaptation. Although we do not have a dataset with images of real scenes in various lighting conditions, it can be reasonably emulated by multiplying images from CIFAR-10 dataset pixelwise by a function $I(x, y)$, which changes between zero and one. Patch normalization discards the overall normalization constant in every patch, which is strongly dependent on the light conditions, and focuses chiefly on the shape of an object that a given unit sees.

Examples of images constructed this way are shown in Fig. 3, where $\approx 80\%$ of each image was covered by a shadow having $I(x, y) = 0.3$. Human can see and correctly classify these "shadowed" images. Two networks: standard CONV net trained end-to-end, and NNL-CONV net trained as described in section 3 were trained on raw CIFAR-10 images, but tested on the shadowed images. Both networks had exactly the same architecture, shown in Fig. 2 (middle). The results are shown in Fig. 3 (right). While the errors on the raw CIFAR-10 images are approximately the same for the two networks ($\approx 23\%$), the error on the shadowed images of the NNL-CONV net is much lower ($\approx 28\%$) compared to the error of the standard CONV net ( more than $\approx 50\%$). This illustrates that patch normalization can be a useful tool for dealing with images having large differences in light intensity (for example coming from shadows), without having images with these kinds of shadows in the training set.

Patch normalization is similar to divisive normalization of [15]. It has also been used in networks trained end-to-end on a supervised task [16].

# 6    Ablations

The networks proposed in this paper include three ideas: unsupervised learning algorithm for the first layer filters, patch normalization of the inputs, and steep non-linearities of the activation functions. Thus, the question remains which of these three ideas (or their combinations) are responsible for the overall performance of the proposed networks. In order to answer this question a series of experiments was performed eliminating each of these three ideas from the final network.

The classification accuracy of the network shown in Fig.2 middle was benchmarked by varying powers of the activation functions, and including/excluding patch normalization. The results are shown in the table below. In this set of experiments the powers of the activation functions in all five blocks were the same and equal to $n$, as specified in the table below.

| Local Learning | | | Backpropagation | | |
|---|---|---|---|---|---|
| power $n$ | patch normalization | test error | power $n$ | patch normalization | test error |
| $n = 1$ | YES | 44.43% | $n = 1$ | YES | 26.93% |
| $n = 10$ | YES | 26.47% | $n = 3$ | YES | 28.39% |
| $n = 40$ | YES | 23.56% | $n = 5$ | YES | $\approx 90\%$ |
| $n = 1$ | NO | 46.92% | $n = 1$ | NO | 22.57% |
| $n = 10$ | NO | 66.94% | $n = 5$ | NO | 33.68% |
| $n = 40$ | NO | $\approx 90\%$ | $n = 10$ | NO | 39.87% |

Thus, the best performance of the local learning algorithm is achieved when the power is high ($n = 40$) and the inputs are patch normalized. The best performance of the backpropagation algorithm is observed when the power is small ($n = 1$) and no patch normalization is used (this is the vanilla CNN). Backpropagation algorithm fails to optimize the loss function for $n \geq 5$ if patch normalization is used, resulting in both training and test errors $\approx 90\%$. The test error also increases as the power $n$ is increased with and without patch normalization. Local learning without patch normalization for large $n$ results in huge numbers as inputs to the classifier and fails to optimize too. With patch normalization the accuracy of the local learning algorithm improves as the power $n$ is increased. This trend, however, reverses when the powers are too big $n \geq 80$. That's why there is an optimal set of powers reported in Fig. 2.

These experiments suggest that both patch normalization and the use of steep non-linearities are absolutely crucial for the good performance of our algorithm. Conversely, it is very difficult to train neural networks with steep non-linearities, like the ones considered in this study, using standard optimizers based on the backpropagation algorithm.

## Acknowledgements

We thank Quanfu Fan, Hilde Kuehne, and Chay Ryali for useful discussions.

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
