# OpenReview forum: "Local Unsupervised Learning for Image Analysis"
_NeurIPS.cc/2019/Workshop/Neuro_AI — Real Neurons & Hidden Units @ NeurIPS 2019 Poster_

### Official Review · AnonReviewer1 · 2019-09-26
**Unsupervised feature learning offers robustness to local shadows**

**Clarity:** 4

**Comment:**

I would be interested to see how the learning rules work when stacked.

**Category:**

Common question to both AI & Neuro

**Clarity Comment:**

Please discuss the relationship of the unsupervised learning rules to the neocognitron.

**Evaluation:**

4: Very good

**Importance:**

4: Very important

**Importance Comment:**

An interesting and relevant study for the workshop. Offers important insights from vision neuroscience that can have specific and concrete impact for DL approaches.

**Intersection:**

5: Outstanding

**Intersection Comment:**

See importance

**Rigor Comment:**

no issue to raise.

**Technical Rigor:**

4: Very convincing

---

### Official Review · AnonReviewer3 · 2019-09-26
**Minor variations to an already published algorithm. Experimental results are suggestive, but limited.**

**Clarity:** 2

**Comment:**

This paper presents several experiments with an algorithm that is likely of interest to attendees of the workshop, but that has been published elsewhere. The experimental results presented in this paper are interesting, but very limited and hard to evaluate. The paper makes relatively strong claims it does not support empirically.

**Category:**

Common question to both AI & Neuro

**Clarity Comment:**

The paper reads well. But unfortunately it (i) includes no details of the algorithm used and (ii) gives no intuition for why this algorithm should be expected to be competitive with backpropagation, and (iii) overstates the strength of its results. Regarding the third point: the paper claims to present evidence that local Hebbian learning is competitive with backpropagation. The evidence they present for this is that the algorithm in [1] can learn convolutional filters and that it can outperform a simpler, shallow network trained with backprop on CIFAR-10. This evidence misses the point of claims of the benefits of backpropagation, which are most clear in the context of deep networks (not networks with single hidden layers) and on real-world or large-scale tasks.

**Evaluation:**

2: Poor

**Importance:**

2: Marginally important

**Importance Comment:**

This paper presents experiments to evaluate the performance of a biologically plausible unsupervised learning algorithm (presented in [1]), a topic of interest to the audience of this workshop. However, the work as presented here is limited. The algorithm is not explained here, making the work difficult to understand. The experiments, while suggestive, are limited and have confounds that make them difficult to interpret. Additionally, the paper appears to overstate its results.

**Intersection:**

5: Outstanding

**Intersection Comment:**

This work presents additional results for an unsupervised learning algorithm that could plausibly be implemented by a biological neural system. As such, the topic it addresses is of interest to both AI and neuroscience communities.

**Rigor Comment:**

The evaluation is limited:
(i) The paper claims to show that Hebbian learning is competitive with backpropagation. But it does not evaluate any deep neural networks, where this claim is typically applied.
(ii) The paper presents learned convolutional filters to demonstrate the strength of their algorithm. These filters are evocative, but it's unclear what they imply about the function learned by the network. The presence of filters like these is at best a sanity check, not a demonstration that the network is competitive with backpropagation.
(iii) Many other learning methods that do not use backpropagation have been demonstrated to learn convolutional filters. I'm mostly aware of results on grayscale images (for methods including independent component analysis and sparse coding), but this likely holds on RGB images as well. This isn't discussed in the paper and no comparisons to other methods that don't use backprop are given.
(iv) The results shown in Figure 3 suggest that using "patch normalization" (i.e. scaling the output of a layer to have norm 1) or using an additional power nonlinearity give robustness to contrast changes in a patch at test time. This is an interesting result, but it is not central to the paper's claim. From the results, it's not clear if this is primarily due to the normalization applied to patches or to the nonstandard nonlinearity used. No ablations are presented. It's unclear how general this apparent robustness is: does it hold for other types of image distortions, or just the particular shadowing presented?
(v) Arguably the main result of the paper, presented in figure 2, shows that a network trained with their algorithm performs similarly to backpropagation. However, the network trained with backpropagation uses a simpler nonlinearity (ReLU) and no patch normalization. For this comparison to be fair, the two networks should be trained with the same architecture. It is likely that the nonlinearity used for the Hebbian model (but not the backprop model) is better suited for this task, given the results shown in Figure 3 (discussed above). As such, it is not clear that the Hebbian learning algorithm is the source of the source of the model's performance in Figure 2.
(vi) The paper reports test errors on CIFAR-10 of ~22%. These are not competitive numbers on CIFAR-10 (errors <10% are standard). This is not surprising given that only single-hidden-layer networks are presented here, but this result should be contextualized.

**Technical Rigor:**

2: Marginally convincing

---

### Official Review · AnonReviewer2 · 2019-09-27
**Interesting results, more details would be helpful**

**Clarity:** 3

**Comment:**

More details are needed, more clear differentiation from previous work, and more explanation of how this is a general approach to the grand problems discussed in the intro, or if it has limited applicability to a specific problem.

**Category:**

Neuro->AI

**Clarity Comment:**

The writing is fine. Again, lack of details affects the overall clarity, however.

**Evaluation:**

3: Good

**Importance:**

3: Important

**Importance Comment:**

This work could be important. Two issues: the work is billed as a general advance, but the approach is designed for and tested on a very specific problem. Second, their are not enough details presented to evaluate the specifics of the algorithm and differentiate the work from previous work.

**Intersection:**

4: High

**Intersection Comment:**

Highly relevant to neuro/ai

**Rigor Comment:**

The work may be quite technically sound. Not enough details are presented to evaluate.

**Technical Rigor:**

2: Marginally convincing

---

### Decision · Program_Chairs · 2019-10-02

Accept (Poster)